# Zoonotic Cycle of American Trypanosomiasis in an Endemic Region of the Argentine Chaco, Factors That Influenced a Paradigm Shift

**DOI:** 10.3390/insects15070471

**Published:** 2024-06-25

**Authors:** Andrea Gómez-Bravo, Sebastián Cirignoli, Diana Wehrendt, Alejandro Schijman, Cielo M. León, María Flores-Chaves, Javier Nieto, Troy J. Kieran, Marcelo Abril, Felipe Guhl

**Affiliations:** 1Fundación Mundo Sano, Buenos Aires C1061ABC, Argentina; agomez@mundosano.org (A.G.-B.); maria.flores@mundosano.org (M.F.-C.); mabril@mundosano.org (M.A.); 2Centro de Investigaciones del Bosque Atlántico, Puerto Iguazú N3370AIA, Argentina; sebaciri@gmail.com; 3Administración de Parques Nacionales, Parque Nacional Iberá, Mercedes W3470, Argentina; 4Instituto de Investigaciones en Ingeniería Genética y Biología Molecular “Dr. Héctor N. Torres”, Buenos Aires C1428ADN, Argentina; dwehrendt@gmail.com (D.W.); schijmanster@gmail.com (A.S.); 5Centro de Investigaciones en Microbiología y Parasitología Tropical, Universidad de los Andes, Bogotá 111711, Colombia; cm.leon@uniandes.edu.co; 6Unidad de Leishmaniasis y Enfermedad de Chagas, Centro Nacional de Microbiología, Instituto de Salud Carlos III, 28222 Majadahonda, Spain; jfnieto@isciii.es; 7Department of Environmental Health Science, College of Public Health, University of Georgia, Athens, GA 30602, USA; tkieran@cdc.gov

**Keywords:** Chagas disease, *Trypanosoma cruzi*, zoonosis, spillover, triatomines, human intervention, vector control

## Abstract

**Simple Summary:**

Chagas disease (*American trypanosomiasis*) poses a serious health problem in the American region, with approximately one-quarter of the Latin American population at risk of infection due to the geographical distribution of the insect vectors (*Triatominae* spp.). The transmission scenarios for this disease involve multiple interdependent factors. The parasite has developed, over evolutionary time, multiple strategies that have enabled its successful survival. The variables governing the transmission cycle are diverse and unique to each ecological scenario. Human intervention, such as deforestation, large-scale changes in land use, and loss of biodiversity, in the Chaco region over many years, along with sustained control interventions, may have significantly impacted the structure of wild transmission cycles and their relationships with the environment. This could potentially reduce the prevalence of the parasite in the domestic cycle. This study aims to describe the transmission dynamics of the sylvatic cycle of *T. cruzi*, identify factors determining potential zoonotic spillover, and explore how they could contribute to the elimination of Chagas disease as a public health problem in this area of Argentina.

**Abstract:**

*Trypanosoma cruzi*, the causative agent of Chagas disease (*American trypanosomiasis*), is a highly complex zoonosis that is present throughout South America, Central America, and Mexico. The transmission of this disease is influenced by various factors, including human activities like deforestation and land use changes, which may have altered the natural transmission cycles and their connection to the environment. In this study conducted in the Argentine Chaco region, we examined the transmission dynamics of *T. cruzi* by collecting blood samples from wild and domestic animals, as well as triatomine bugs from human dwellings, across five sites of varying anthropic intervention. Samples were analyzed for *T. cruzi* infection via qPCR, and we additionally examined triatomines for bloodmeal analysis via NGS amplicon sequencing. Our analysis revealed a 15.3% infection rate among 20 wild species (n = 123) and no *T. cruzi* presence in 9 species of domestic animals (n = 1359) or collected triatomines via qPCR. Additionally, we found chicken (34.28%), human (21.59%), and goat (19.36%) as the predominant bloodmeal sources across all sites. These findings suggest that anthropic intervention and other variables analyzed may have directly impacted the spillover dynamics of *T. cruzi*’s sylvatic cycle and potentially reduced its prevalence in human habitats.

## 1. Introduction

Chagas disease (*American trypanosomiasis*) is a very complex zoonosis that is present throughout South America, Central America, and Mexico and continues to represent a serious threat to the health of 21 endemic countries in the region. An estimated 6 million people are infected with the causative agent, the protozoan parasite *T. cruzi* (Kinetoplastida: Trypanosomatidae), causing 12 thousand annual deaths and approximately 9000 newborn infections during gestation, with a further 75 million people, roughly one-quarter of the Latin American population, at risk of infection due to the geographical distribution of the insect vectors [1]. There are several modes of transmission of *T. cruzi*; traditionally, the most important is through the vector route, with different species of hemipteran insects of the Reduviidae family acting as natural vectors both in the sylvatic and domestic cycles. Other infection routes include blood and organ donation, oral transmission through the ingestion of contaminated food, and the secretions of some mammalian animal reservoirs.

Currently, 154 species within the *Triatominae* subfamily (three extinct and 151 still present) and the vast majority have been described as capable of transmitting *T. cruzi* under natural and experimental conditions [2]. Most species occupy sylvatic ecotopes in association with their respective vertebrate hosts. More than 180 species of terrestrial and arboreal mammals, 7 orders, and 25 subfamilies have been reported to be infected with the parasite in natural environments. Although all mammals are considered capable of being infected with *T. cruzi*, the groups of marsupials, armadillos, carnivores, and rodents are the main wild reservoirs [3]. *T. cruzi* has established long-lasting infections in its mammalian hosts as a product of evolutionary balance achieved through evolutionary time, with dynamic interaction patterns that play different roles in the transmission network of the parasite in the variety of habitats and biomes where it is found [4]. Examples include palm crowns, bird nests, opossum lodges, rock-piles, hollow trees, rodent nests, and bat caves. In most cases, triatomine species are exquisitely adapted to their ecotopes, with little propensity to invade human habitations. Therefore, there are approximately 10 to 15 species of triatomines that show anthropophilic tendencies and are regularly implicated in parasite transmission to humans [5]. The process of domestication of populations of triatomine species undergoes simplification given their adaptation to the stable domestic environment, resulting in the evolution of domestic genotypes that would have reduced fitness in sylvatic ecotopes [6]. This has proven to be the case of the species *Triatoma infestans*, which until recently had a large distribution across Argentina and was present in all the countries in the southern cone [1]. Other species, such as *Triatoma guasayana,* present an abundant population in the El Chaco region, mainly in biotopes that include Cactaceae and logs, feeding on rodents, marsupials, and birds. *Triatoma garciabesi* populations are also present in the region, mainly associated with roosting chickens, and chicken coops. These last two species are considered secondary vectors and are associated with the peridomicile [7,8,9].

The transmission of *T. cruzi* may be interrupted by residual insecticide spraying to kill the domiciliated triatomine vectors and for the other routes of transmission, such as mother-to-child through vertical transmission. Despite extensive control programs, the disease remains a serious obstacle to health and economic development in Latin America, especially for the rural poor, and very little is known about the epidemiological situation in socially and economically depressed endemic areas [10].

Vector-borne transmission of Chagas disease in rural dwellings is proximally determined by housing conditions and the management of domestic animals, and the repertoire of control measures is limited. Although there are some vaccines in development, none have been approved for human use yet. Currently, two drugs are the only arsenal available for treatment, and they are more effective during the acute and early chronic phases of the disease [11]. 

In 1991, the governments of six of the most heavily affected countries (Argentina, Brazil, Bolivia, Chile, Paraguay, and Uruguay) in conjunction with the PAHO/WHO set up an intergovernmental control program known as the ‘Southern Cone Initiative’, designed to halt Chagas disease transmission by eliminating its main insect vector, *T. infestans*, and universal screening of blood donors [1]. With the steady progress of control and prevention strategies, the Chaco region of Argentina is now one of the last frontiers of the Americas to receive concerted action against Chagas disease. 

In the study area, several approaches to Chagas disease surveillance and control have been implemented, such as insecticide spraying, the improvement of houses and peridomestic structures with community participation, avoiding cracks in walls and ceilings, which are the main refuges for vector insects, primary health education programs in the community, diagnosis and treatment in the infected population without risk of reinfection due to vector transmission, and epidemiological surveillance techniques [12,13]. Despite control activities in the domestic habitat, given the zoonotic nature of this disease, elimination as a public health problem is the target, as stated in the new WHO road map for 2023 [14].

Eradication of this disease is not considered due to the possibility of zoonotic spillover from the sylvatic to the domestic cycle. Nonetheless, zoonotic spillover requires the conjunction of several factors, including ecological, epidemiological, and behavioral determinants of pathogen exposure and the within-human factors that affect susceptibility to infection [15]. Thus, there are numerous questions about the epidemiological role played by many host species, mainly due to the complexity of the processes and the ecological interrelationships.

One of the factors that has most influenced the transmission scenario of *T. cruzi* in the studied region, in addition to the sustained control activities conducted since 2005 [16], is the impact of human activities in the area. Changes in land use, the expansion of the agricultural frontier, and the indiscriminate hunting of wild animals have greatly modified the environment surrounding these rural settlements [17]. All these factors have probably affected the dynamics of transmission in this area and influenced the connection between the sylvatic and domestic habitats. Therefore, in this study, we aim to describe the transmission dynamics of the sylvatic cycle of *T. cruzi* to determine the potential zoonotic spillover that might hamper the elimination of Chagas disease as a public health problem in this area of Argentina.

## 2. Materials and Methods

### 2.1. Study Area

The study was carried out in rural settlements located in the departments of General Taboada and Juan Felipe Ibarra, province of Santiago del Estero, Argentina. The selected areas are located within the semi-arid Chaco region, and the rural settlements consist of small groups of houses scattered throughout the territory (Figure 1). 

The houses are built mainly with adobe (either with mud directly or with adobe bricks), wood, plastics, sticks, and thatched roofs. Around each dwelling, peridomestic structures are built for the confinement of animals and food storage [16]. The population is supported by subsistence economies that involve the use of wood, wildlife, and extensive livestock, mainly goats, which usually graze freely within the forest matrix [18]. 

The landscape is characterized by the presence of patches of secondary forest with a predominance of timber species that have been heavily exploited since the early 1920s to feed the tannin industry, as charcoal, and as material for railroad tracks [19]. The climate in the region is markedly seasonal, with a cold, dry period and average minimum temperatures of 5 °C (April to September) and a hot, rainy period with average temperatures of 34 °C and a maximum of 50 °C (October to March). Rainfall reaches average values of 740 mm per year.

### 2.2. Sampling Sites

Five sampling sites were selected according to their degree of anthropic intervention and based on the infrastructure in each area:No intervention: Adobe Natural Reserve (ANR), a lack of roads, constructions, and deforestationSemi-intervened: Lote 59 (L59) and El Desvío (ED), a few dirt roads, constructions, and deforestationIntervened: Miel de Palo (MP) and Malacara (M), with paved roads, constructions, and deforestation. 

The control and surveillance activities were carried out at the sites with different degrees of intensity and times, including insecticide spraying, sanitary improvement of human habitat (pens, cisterns, latrine, toilet, internal and external plaster, ceilings, and external painting), and diagnostic and treatment operations (Table 1).

### 2.3. Ethical Considerations and Biosecurity Measures

The manipulation of animals and the collection of blood samples from different wild and domestic animals were ethically approved by the General Directorate of Forests and Fauna of the province of Santiago del Estero (Resolution No. 1481). Sampling of domestic animals in all the dwellings in each of the settlements and informed consent from each of the owners were obtained. The handling and sampling of these animals were carried out by veterinarians, with the active help of the members of each family, following the protocol of animal integrity and welfare.

For biosecurity measures, all field personnel involved in the handling of animals received personal protection equipment composed of clothing, shoes, gloves, face masks, and security glasses as indicated by protocol [20]. Additionally, recommendations of vaccination for rabies prophylaxis and booster shots for tetanus and diphtheria were followed, as per the Department of Rural Zoonosis from the Ministry of Health of Buenos Aires province. 

### 2.4. Capture and Sampling of Wild Mammals

Animals were captured and blood samples collected in 2017 during the winter months (July/September) and in the summer and early fall months of 2018 (February/April). Winter is considered the dry season, while the months of February to April are the rainy season. Each capture activity for sampling was carried out for 10 consecutive days at each site. For the intervened areas, a network-type catching design was implemented, a scheme adapted from Parmenter et al., 2003 [21]. Twelve linear transects of 100 m with 18 capture stations were installed, covering an area of 3.14 ha with a total of 216 traps: 56 Tomahawk and 160 Sherman. 

In each of the 10 transects (Figure 2), 5 Tomahawk and 13 Sherman traps were placed with distances of 5 m for the [Sherman–Sherman] and [Sherman–Tomahawk] combinations, and 10 m for the [Tomahawk–Tomahawk] combinations. In the remaining 2 transects, 3 Tomahawks and 15 Shermans were placed according to the availability of total traps. Each station was georeferenced (Garmin Etrex, Olathe, KS, USA) and coded for control and review purposes.

For the non-intervened site of the ANR, the park authorities did not allow the use of a network design to avoid disturbing the natural flora and fauna; therefore, capture stations were installed steps away from pre-existing paths. Tomahawk traps were randomly baited with bacon, beef, chicken offal, fish, and fruit. Sherman traps were baited with pellets made with peanut butter, oatmeal, vanilla essence, and chocolate. All catching stations were checked twice a day, in the morning and before sunset, and rebaited if necessary [22]. The capture effort for non-flying mammals was calculated by multiplying the number of traps used by the number of sampling nights.

For bat captures, three mist nets (one 12 m × 2.5 m and two 6 m × 2.5 m with 38.0 mm black mesh; AFO Banding Supplies, Manomet, MA, USA) were placed in open areas within the forest. The nets were placed after sunset and monitored every 30 min for a period of 4 h. Additionally, in MP, bats from a colony found in the attic of a house were sampled, implementing a homemade trap (“bin trap”) that was placed covering the refuge exit, while three other bats were captured at L59 from an old water well using a homemade net. The capture effort for bats was calculated by multiplying the number of lineal meters of net used each night by the total number of hours the nets were open. 

All trapped animal specimens were given parenteral anesthesia for induction composed of Zolazepam hydrochloride (Zelazol^®^, Zoetis Argentina, Villa Adelina, Argentina) or 10% Xylazine (Ronpun^®^, Bayer Leverkusen, Alemania) and 5% Ketamine hydrochloride (Vetaset^®^, Vetsmart, San Pablo, Brasil) injected intramuscularly, depending on the species and weight [23]. Animals were bled by venipuncture, and each blood sample was separated into three aliquots. Once they were sedated, all animals were characterized and identified according to Barquez & Díaz (2009) [24], weighed with spring scales, and standard body measurements were recorded, including body length, tail length, ear length, and hind leg length. Additionally, all captured animals were marked by shaving an area of the body and released in the same place of capture, except for mice and bats that were sacrificed for taxonomic determination. These specimens were identified in the Bernardino Rivadavia Argentine Museum of Natural Sciences (MACN) and donated to the Mastozoological Collection (Appendix A).

### 2.5. Sampling of Domestic Mammals

A domestic animal census was carried out in all study areas to estimate a representative number of animals to be sampled. Census data included species, sex, and age (<1 year, ≥1 year). According to these numbers and their role in the trypanosomiasis cycle, the following percentage of the population was sampled: 100% of dogs and donkeys, 50% of all cats, pigs, and mules; 25% of all cows and horses; 20% of all sheep; and 10% of all goats. 

Blood samples collected from both wild and domestic mammals were kept in guanidine hydrochloride-EDTA buffer (GEB) in a 3:1 ratio at room temperature in the field laboratory until they were sent to the Instituto de Investigaciones en Ingeniería Genética y Biología Molecular (INGEBI) in Buenos Aires (Argentina) for analysis to determine the presence of *T. cruzi*. Aliquots of a subset of samples from dogs and wild animals were also sent to the Centro Nacional de Microbiología del Instituto de Salud Carlos III (CNM) in Madrid (Spain) for molecular analysis (double blind way). 

### 2.6. Molecular Diagnosis of T. cruzi in Mammals

In the INGEBI laboratory, DNA extraction was performed using a commercial kit, the High Pure PCR Template Preparation Kit (Hoffman-La Roche Ltd., Basel, Switzerland). A duplex real-time PCR was developed with Taqman probes for the simultaneous detection of *T. cruzi* satellite DNA and a conserved endogenous gene from mammals, the interphotoreceptor retinoid-binding protein (IRBP) [25].

In the CNM laboratory, DNA extraction was performed using an automatic system (QIAsymphony SP, QUIAGEN, Hilden, Germany) and the QIAsymphony^®^ DNA Midi kit (QIAGEN, Hilden, Germany). Coagulated samples were previously lysed with the addition of 1 mL of lysis solution composed of 620 µL of NET 10, 375 µL of 10% SDS, and 5 µL of 50 µg/µL of Proteinase K in proportion to the volume of guanidine in each sample. The tubes were then incubated in agitation at 56 °C overnight. In each step of the DNA extraction protocol, positive and negative controls were added. Subsequently, a multiplexed real-time PCR based on the amplification of the variable region of kDNA and satellite DNA was performed. This PCR amplifies the variable region of the kDNA minicircle using 121 and 122 primers [26] and the satellite repetitive sequence using Tc1 and Tc2 primers [27]. To evaluate the presence of inhibitors, a conserved sequence of the mammalian gene from the ribosomal subunit 18S was used [28].

For verification, blood samples from domestic dogs were also analyzed using the commercial kit Chembio Chagas STAT PAK^®^ Assay as per the manufacturer’s instructions and positive and negative controls from a previous study [29]. These samples were processed at the Centro de Investigaciones en Parasitología Tropical (CIMPAT), Universidad de los Andes, Bogotá, Colombia.

### 2.7. Capture and Sampling of Triatomines

Angulo-type cage traps were used with a live chicken as bait. These types of traps allow the collection of live triatomines, are easy to transport, and avoid the intervention of habitats, in addition to being more effective than other similar traps [30]. During the winter sampling, 5 traps were randomly distributed at each wild animal sampling site within the same delimited area. Traps were placed in potential triatomine habitats such as forks in the canopy of trees, hollow tree trunks, roots, and fallen trunks, as well as in the entrance of animal burrows and on the soil, and placed for 6 consecutive nights, installed at dusk, and removed in the morning for the collection of any captured triatomines. 

During the summer sampling, traps were installed in the peridomicile, in structures such as goat pens, pigs, and chicken coops, and behind or under different objects located around the houses, such as firewood, bricks, and corn deposits, among others. These traps were installed for 9 consecutive nights and, as in the winter sampling, installed before sunset and removed in the morning for the collection of any triatomine. 

All captured triatomines were stored in containers for hemoculture and identified with a locality code and date of capture. Once in the lab, each specimen was characterized using morphological keys [31]. Adult and 5-instar nymph specimens were processed by compression of the abdomen to determine the presence of trypanosomatids in their feces. Feces were diluted in a 0.9% saline solution, placed between a slide and coverslip, and observed under an optical microscope. All the triatomine samples collected during the study were conserved at 4 to 5 °C in the field laboratory and then transported to INGEBI for their analysis. In addition, a subset of samples was sent to the Glenn Lab (Dept. of Environmental Health Science, University of Georgia, United States, for detection of *T. cruzi* and blood meal analysis. 

### 2.8. Molecular Diagnosis of T. cruzi in Triatomine Insects

For the extraction of samples conserved in alcohol, tissue from the posterior end of the abdomen was placed in a microtube with 100 µL of PBS, which was conserved at 4 °C until extraction. The extraction was performed in INGEBI using a commercial DNA extraction kit (either QIAGEN or Roche), following the same protocol recommended for mammalian tissue but extending the lysis reaction for 24 h. Purified DNA was conserved at −20 °C until further use. 

For the extraction of DNA from filter paper, a slight modification of the methodology described by Machado et al. [32] was used. Briefly, a 6 to 10 mm square portion of the filter paper with the sample on it was placed in a microtube, and 200 µL of distilled H2O was added. This was incubated at 96 °C for 15 min, then left to cool at room temperature and centrifuged at 14,600 rpm for 5 min [33]. The supernatant was conserved at −20 °C until further use [34]. For these samples, the following inhibitory controls were performed: (1) *T. cruzi* DNA was added (5 µL of 100 eq.par/mL DNA) to a duplicate of the sample before the extraction to verify if it was amplified in the qPCR (inhibition control). If there are inhibitors, then no amplification would occur. (2) A total of 200 pg of exogenous DNA were added (IAC: internal amplification control, DNA from a p-Zero plasmid that contains a sequence of *Arabidopsis thaliana*) to the sample before DNA extraction to see if it’s amplified by the qPCR. The IAC is usually used in the detection of *T. cruzi* from human samples [35]. (3) Known quantities of epimastigotes from the CL-Brener strain were added to a sample before extraction. For this, the filter paper containing the sample was cut into equal parts, adding in each part 10, 20, 50, and 100 parasites, respectively. 

A duplex qPCR for the simultaneous detection of satellite DNA of *T. cruzi* and the 12SrRNA of triatomines as an internal control (SatDNA/Triat) was used with the primers cruzi1c and 2c (750 nM each) and the probe TaqMan cruzi3 (50 nM) [33], combined with 300 nM and 500 nM of primers P2B and P6R, respectively, and 50 nM of probe Triat [34]. The PCR was performed in a final volume of 20 µL using 2 µL of DNA extracted from the triatomine samples, using the following cycling on either an ABI 7500 thermocycler (Applied Biosystems^®^, Waltham, MA, USA): 10 min at 95 °C, followed by 40 15-s cycles at 95 °C, and finally 1 min at 58 °C. In CIMPAT, a total of 123 triatomines (12 *T. garciabesi*, 23 *T. guasayana*, and 88 *T. infestans*) were subjected to DNA extraction from gut samples following the protocol by Kieran et al., 2017 [36]. Subsequently, macerated samples were placed in a digestion buffer with Proteinase K and digested for two hours at 55 °C before extraction with phenol-chloroform-isoamyl alcohol (PCI). The extracted DNA was reconstituted in 30 µL Tris-Low-EDTA buffer (10 mM Tris, pH 8, 0.1 mM EDTA). For detection of *T. cruzi* DNA, TaqMan^®^ Universal PCR Master Mix (Applied Biosystems), Cruzi1/Cruzi2 primers, and Cruzi3 probe (FAM/NFQ-MGB Applied) were used in a final reaction volume of 20 μL. Reactions were run in 96-well plates using an ABI 7500 FAST thermocycler (Applied Biosystems^®^). 

### 2.9. Bloodmeals in Triatomines

The extracted DNA from the 123 triatomines was amplified for vertebrate 12S rRNA following [36]. We used 12S primers from Humair et al. 2007 (F-5′-CAA ACT GGG ATT AGA TAC C-3′, R-5′-AGA ACA GGC TCC TCT AG-3′), for which we attached the Illumina TruSeq adaptor sequence and unique 8 + 12 bp barcodes onto each respective forward and reverse primer for 96 inner dual-index combinations [36,37]. We then performed replicate PCRs using KAPA HiFi Hotstart (KAPA Biosystems, Wilmington, MA, USA) using the following thermocycler conditions: 98 °C for 3 min, followed by 40 cycles at 95 °C for 45 s, 63 °C for 1 min, 72 °C for 30 s, and a final extension at 72 °C for 1 min. Positive amplification was verified on a 1.5% agarose gel, and each sample in 96-well plates was pooled in equal molar amounts and cleaned with a 2:1 ratio of SeraMag Speedbeads, reconstituting in TLE. A second round of PCR in triplicate was then performed on each pool to make full Illumina compatible libraries with outer barcodes [38] using 98 °C for 2 min, followed by 6 cycles at 98 °C for 30 s, 60 °C for 30 s, 72 °C for 30 s, and a final extension at 72 °C for 5 min. Second-round libraries were cleaned with speadbeads in a 2:1 ratio and pooled for paired-end sequencing on an Illumina MiSeq v3 600 cycle kit at the Georgia Genomics and Bioinformatics Core (Athens, GA, USA). Sequencing data was demultiplexed by outer barcodes on Illumina BaseSpace and then by internal barcodes with Mr. Demuxy v1.2.0 (https://pypi.python.org/pypi/Mr_Demuxy/1.2.0, accessed on 17 June 2024). We then imported the data into Geneious Prime version 2019.1.1 (bioMATTERSed, New York, NY, USA) to pair, merge, and trim the reads of primers and low-quality bases. Data were then exported as FASTQ files and imported into QIIME2 [39] for taxonomic classification using a custom reference database and taxonomy file of relevant sequences from Genbank [36]. We used DADA2 [40] to dereplicate and remove chimeras, followed by taxonomic classification using sklearn.

## 3. Results

### 3.1. Wild Animals

A total of 129 specimens of wild mammals were captured belonged to 20 species from seven orders: Didelphimorphia, Cingulata, Carnivora, Cetartiodactyla, Chiroptera, Rodentia, and Lagomorpha. Two species corresponded to exotic invasive species, one with domestic/peridomestic habitats (*Rattus rattus*) and a Leporidae widely distributed throughout the country (*Lepus europaeus*) (Table 2).

The capture effort for non-flying mammals was 21 nights per trap (10,800 nights per season), of which 5.600 nights/traps correspond to Sherman traps. While the capture effort for Chiroptera was 960 h/net. 

In the traps, there was a predominance of skunks (*Conepatus chinga*), a generalist species that tolerates intervened environments. A great percentage of terrestrial mammals were captured by local inhabitants, who mainly use these species for consumption and, to a lesser degree, as pets. Another terrestrial species that was abundant with respect to the rest was the armadillo, *Chaetophractus villosus*. The flying mammals were mostly captured through the “bin trap”, and the highest abundance was composed of three species that used the ceiling of a household as a refuge: *Promops nasutus*, *Eptesicus* cf. *furinalis*, and *Myotis* sp. Species richness was slightly higher in sites with anthropogenic disturbances, such as M (10 species) and MP (8 species), in comparison to the non-intervened ANR (6 species) and the semi-intervened ED (5 species).

### 3.2. Domestic Animals

A total of 7769 individuals belonging to nine different species were recorded in the census performed in the study area. Representative samples of each species were sampled as mentioned in the Section 2, reaching a total of 1388 sampled animals (Table 3). The proportion of domestic animal samples shows the predominance of *Capra aegagrus hircus* (goat) in the area and a high number of *Canis lupus familiaris* (n = 357). 

### 3.3. Triatomines

A total of 243 triatomines were captured during inspection of the peridomicile (animal pens, chicken coops, piles of bricks, deposits, etc.) in each of the sampling sites. Only three species, *T. infestans* (n = 204; 84%), *T. guasayana* (n = 28; 11.5%), and *T. garciabesi* (n = 11; 4.5%), were found. Regarding the capture of triatomines per study site, the highest number of specimens were captured in M (n = 111; 46%), L59 (n = 70; 29%), MP (n = 32; 13%), and ED (n = 30; 12%).

### 3.4. Detection of Trypanosoma cruzi

From the total of blood samples collected from wild mammals (n = 129), 123 were processed using the duplex qPCR cruzi/IRBP performed in the INGEBI laboratory. This method worked well for almost all the species of wild animals sampled, except for *Chaetophractus villosus*, *Tolypeutes matacus*, *Myotis* spp., *Eptesicus* cf. *furinalis,* and *Graomys chacoensis*. Given that the qPCR cruzi/IRBP did not amplify well in the samples from these species, a beta-actin qPCR was performed. Most of the samples (94 out of 123; 76.4%) had no detectable *T. cruzi* DNA. The remaining 29 samples were indeterminate given that the qPCR cruzi/IRBP did not amplify and neither did the PCR with the beta-actin gene. In the CNM laboratory, 124 of the 129 samples from wild animals were processed using kDNA multiplex real-time PCR, with 19 (15.3%) having detectable *T. cruzi* DNA for one of the target genes. In the remaining 105 samples, there was no trace of *T. cruzi* DNA. 

With respect to the analysis of the samples from domestic animals, the qPCR cruzi/IRBP performed by the INGEBI laboratory worked well for all the species, and there was no detectable *T. cruzi* DNA in any of the 1359 samples analyzed. The CNM laboratory analyzed 264 of the dog samples collected using *T. cruzi* multiplex real-time PCR, but there was no detection of *T. cruzi* DNA. To confirm these results, serological diagnosis was performed on the collected dog samples, and 16 out of 314 (5.1%) of the analyzed samples were positive for *T. cruzi* antibodies.

Analysis for the presence of *T. cruzi* in the triatomine specimens captured showed a lack of parasites or parasite DNA, both by direct observation of adults and 5-intar nymphs under the microscope (n = 35, all undetectable) and by molecular methods from specimens stored whole in alcohol (n = 45) and feces collected in filter paper (n = 80). The molecular analysis of the specimens in alcohol was performed in the INGEBI laboratory using the duplex qPCR SatDNA/Triat, which appropriately amplified the internal control (Ct values: 16.4–24.3) and showed no inhibition. The molecular analysis of the fecal samples in filter paper was also processed using duplex qPCR SatDNA/Triat with positive and negative controls, as well as inhibitory tests, yet *T. cruzi* DNA was not detected in any of the samples. From the 120 specimens of triatomines captured in alcohol during the secondary sampling, 95 were also analyzed for traces of *T. cruzi* DNA, and there was no detection of parasites.

### 3.5. Bloodmeal Results

Overall, 23 taxa were detected in the bloodmeals for all triatomines sampled across sites, with a low, non-significant (*p* = 0.4405, Kruskal–Wallis) difference in alpha diversity between species and sites (AlphaDiversity_Shannon, AlphaDiversity_Simpson). The most abundant taxa averaged across samples were chicken (*Gallus gallus*, 34.28%), human (21.59%), goat (*Capra* sp., 19.36%), pig (*Sus scrofa*, 7.06%), dog (*Canis lupus*, 5.32%), and bat (*Eptesicus* sp., 4.4%). When examining across sites and triatomine species, chicken, human, and goat are predominately in the top three species detected, collectively with >73.74% of the detection (Figure 3). A couple of exceptions to this include *T. guasayana* at L59, where dogs were the majority (50.25%), followed by humans (15.28%), and chickens (10%), and *T. guasayana* at MP, where human DNA was detected at a much lower frequency (6.73%). At ED, bats (*Eptescius* sp.) had the fourth highest frequency at 13.92%. A few interesting findings include the detection of frogs, *Scinax nascius* at ED (8.71%), M (1.5%), L59 (1.12%), MP (1%); and *Physalaemus biligonigerus* at MP (1.5%). We also detected *Didelphidae* sp. (2%) and non-chicken birds (1.5%) across sites.

### 3.6. Serological Diagnostics in Humans

Since 2017, the local health system, in collaboration with Mundo Sano, has been carrying out serological screening on Chagas disease in rural areas without risk of vector transmission. At the sites included in this study (Table 1), between 2017 and 2019, 202 persons were screened (31% coverage), and 68 (34%) were positive with conventional serological procedures: 1 (1.5%) child under 5 years of age, 1 (1.5%) child between 6 and 10 years of age, 1 (1.5%) between 11 and 30 years of age, and 62 adults up to 31 years of age (unpublished data).

## 4. Discussion

*T. cruzi* exhibits a variety of transmission scenarios where many interdependent factors are involved. The variables that influence the *T. cruzi* transmission cycle are diverse and unique to each ecological scenario. The deforestation, large-scale changes in land use, and loss of biodiversity that the Chaco region has experienced may have had a significant impact on the structure and functioning of wild transmission cycles and relationships with the environment, either reducing or increasing the prevalence of parasites in the domestic cycle [43,44,45,46]. Additionally, in the studied areas, sustained control operations have been carried out in recent years, which has led to a drastic decrease in insect vector populations and, consequently, a significant decrease in infection in wild reservoirs and domestic animals. Also, massive deforestation, the growth of transgenic crops with a battery of agrochemicals, and sustained surveillance are generating a negative effect on zoonotic spillover. 

In a study carried out in the Gran Chaco region, Ceballos et al. [43] compared the variation in the transmission cycle of *T. cruzi* between surveys carried out in 1984–1991 and 2002–2004 and observed a significant decrease in the prevalence and incidence over time. In 2002 and 2004, 501 samples of wild mammals from 13 different species were analyzed, and only 2 species were found infected by *T. cruzi*: an opossum, *Didelphis albiventris,* and a skunk, *Conepatus chinga*. These results are consistent with those found by Alvarado-Otegui et al. [47], in which 44 samples of mammals belonging to 14 species were analyzed, of which only two opossums, *Didelphis albiventris,* and one armadillo, *Dasypus novemcinctus,* were infected with *T. cruzi*. In our study, *Didelphis albiventris* was not reported in any of the study sites, although the presence of skunk and armadillo species was reported, none were infected by *T. cruzi.*

The aim of our study is to identify the specific sites affected by human intervention, analyze the consequences of said interventions, and provide conclusions about the possible impacts on the transmission of *T. cruzi* in the Argentine Chaco. The magnitude of the spillover effect will depend on the quality of the habitat as well as its proximity. Interestingly, all the samples of the 129 wild mammals captured in the study areas that belonged to a wide range of taxonomic orders were analyzed for the presence of *T. cruzi*, and the results were negative in a double-blind way in two different laboratories (INGEBI and ISCIII). Two species corresponded to exotic invasive species, one with domestic/peridomestic habitats (*Rattus rattus*) and a Leporidae widely distributed throughout the country (*Lepus europaeus*) (Table 2). There was a predominance of skunks (*Conepatus chinga*), a generalist species that tolerates intervened environments. Another terrestrial species that was abundant with respect to the rest was the armadillo, *Chaetophractus villosus*. The highest abundance of bats was composed of three species that used the ceiling of a household as a refuge: *Promops nasutus*, *Eptesicus* cf. *furinalis*, and *Myotis* sp.

Species richness was slightly higher in sites with anthropogenic disturbances, such as M (10 species) and MP (8 species), showing a predominance of rodents, bats and armadillos in comparison to the non-intervened ANR (6 species) and the semi-intervened ED (5 species), where the presence of carnivorous mammals is greater (Figure 4). According to the dilution effect proposed by Schmidt and Ostfeld, 2001 [48], many infectious diseases in humans are caused by pathogens that reside in nonhuman animal reservoirs and are transmitted to humans via the bite of an arthropod vector. Most vectors feed on a variety of host species that differ dramatically in their reservoir competence, that is, their probability of transmitting the infection from host to vector. Whereby the presence of vertebrate hosts with a low capacity to infect feeding vectors (incompetent reservoirs) dilutes the effect of highly competent reservoirs, thus reducing disease risk. Anthropogenic actions cause a transformation of the landscape and lead to the loss of habitat and biodiversity, resulting in the selection of generalist species that represent a low epidemiological risk depending on their infectious potential. Therefore, a high diversity of species with low potential for infectivity does not constitute a significant risk of transmission of infection, as seems to be the case in the areas studied. Although the results obtained in 19 wild animals are difficult to interpret, the observed levels of *T. cruzi* DNA correspond to low non-quantifiable parasitemias. Different studies have shown that kDNA has the same sensitivity for all genotypes, while the sensitivity of satellite DNA is lower. Because satellite DNA is present in lower copy numbers in the TcI, TcIII, and TcIV genotypes, it is not surprising to find that satellite DNA is negative while kDNA is positive. Discrepancies between the two targets are associated with samples that have non-quantifiable levels of parasitemia, which is precisely what is observed in the results obtained.

Dogs, the main domestic reservoirs of *T. cruzi* in the Argentine Chaco, may be useful as sentinels of vector-mediated transmission of *T. cruzi* in control programs if canine infections acquired by all other routes could be excluded. The presence and proportion of infected dogs in a home are indicators of the presence of infected individuals and a measure of the risk of household transmission [49]. 

By studying dog populations in several localities in the province of Santiago del Estero before and after the fumigation with residual insecticides (in 1992 in Amama, Trinidad, and Mercedes and 1993–1994 in the other villages) and twice during a program following triatomine surveillance (in 1994 and 1996), overall seropositivity for *T. cruzi* infection steadily decreased from 65% to 15% in 1996 [50]. The results obtained in the INGEBI and CNM laboratories by analyzing the samples of 314 dogs out of a total of 357 (95.2%) using *T. cruzi* real-time PCR showed no detection of *T. cruzi* DNA. To confirm these results (double blind way), serological diagnosis was performed in the CIMPAT laboratory, and only 16 out of 314 (5.1%) of the analyzed samples were positive for *T. cruzi* antibodies. Our results agree with those reported by [51,52], demonstrating that sustained control actions are effective and significantly reduce the transmission of the parasite in the human habitat. 

When examining across sites and triatomine species, chicken, human, and goat are predominately in the top three species detected, collectively with >73.74% of the detection (Figure 5). A couple of exceptions to this include *T. guasayana* at L59, where dogs were the majority (50.25%) followed by humans (15.28%), and chickens (10%), and *T. guasayana* at MP, where human DNA was detected at a much lower frequency (6.73%). At ED, bats (*Eptescius* sp.) had the fourth highest frequency at 13.92%. A few interesting findings include the detection of frogs, *Scinax nascius* at ED (8.71%), M (1.5%), L 59 (1.12%), and MP (1%); and *Physalaemus biligonigerus* at MP (1.5%). We also detected *Didelphidae* sp. (2%) and non-chicken birds (1.5%) across sites.

*T. infestans* is a native species and hence widespread in wild environments, across the dry Chaco (Argentina, Paraguay, and Bolivia) and the inter-Andean temperate-dry valleys of south-eastern Bolivia [53]. House reinfestation by native *T. infestans* is common, and transmission of *T. cruzi* can persist or resume (even if at relatively low intensities) in areas under control surveillance [50,54,55,56,57]. *T. garciabesi* and *T. guasayana* displayed a significantly different distribution among peridomestic ecotopes. *T. guasayana* predominated over *T. garciabesi* in the prevalence of infested sites and the number of bugs collected. Our results allow us to predict that *T. guasayana* is responsible for the cases of *T. cruzi* infection observed in dogs at domestic or peridomestic sites during the surveillance program. *T. garciabesi* repeatedly used the rugged bark of *Prosopis alba* or *P. nigra* (Fabaceae) trees, where chickens roosted, and chicken coops. For *T. guasayana,* the main ecotopes were goat or sheep corrals, piled materials, and orchard fences. Neither *T. garciabesi* nor *T. guasayana* colonized human habitations, even in the absence of *T. infestans*, formerly the predominant domestic vector of *T. cruzi* in this area [58].

Additionally, sustained control operations have been carried out in recent years, which has led to a drastic decrease in insect vector populations and consequently a significant decrease in infection in humans, wild reservoirs, and domestic animals. In the region studied, massive deforestation, the growth of transgenic crops with a battery of agroFcals, and sustained surveillance are generating a negative effect on zoonotic spillover.

Disease control needs clear goals, and those goals often heavily depend on the natural history, risk factors, transmission routes and dynamics, pathogenesis, and treatment of the disease [59]. The most ambitious goal is disease eradication, which means the complete elimination of an infection, with no new cases recorded in the absence of control measures. Eradication is practically impossible for zoonoses such as Chagas disease, therefore, the objective of a control program cannot be eradication, at least in the Americas [4,60,61].

Didelphis albiventris is one of the most widely distributed marsupials in Argentina [62] (Chemisquy and Martin, 2019). In the Chaco region, it has been identified as a major reservoir host of *Trypanosoma cruzi* [63,64,65,66,67,68]. Despite its high adaptability to various environmental disturbances and significant sampling efforts, the species was not recorded in the study area. This absence is likely due to the low population density and small size of forest patches, which negatively impact the presence of the opossum as it relies on trees for nesting. Furthermore, indiscriminate hunting in rural areas, where it is perceived as a predator of poultry, exacerbates the decline in its population.

The need for a multidisciplinary approach to the study of health questions has been increasingly recognized. The One Health concept has been gaining impetus, as has the need for tools for the diagnosis of parasitic infections in wild animals [4,61], which opens a new path to understanding the different transmission scenarios of *T. cruzi* in the Americas and thus being able to establish adequate surveillance and control programs.

## Figures and Tables

**Figure 1 insects-15-00471-f001:**
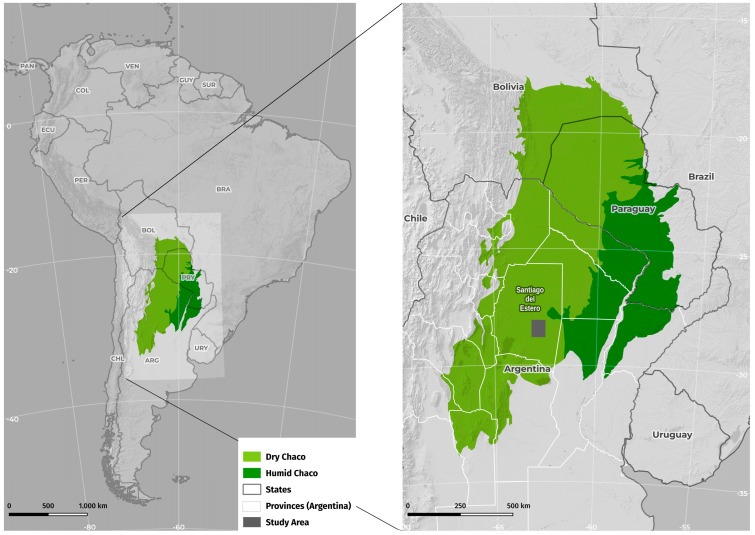
Map showing the location of the study areas in the Chaco Region.

**Figure 2 insects-15-00471-f002:**
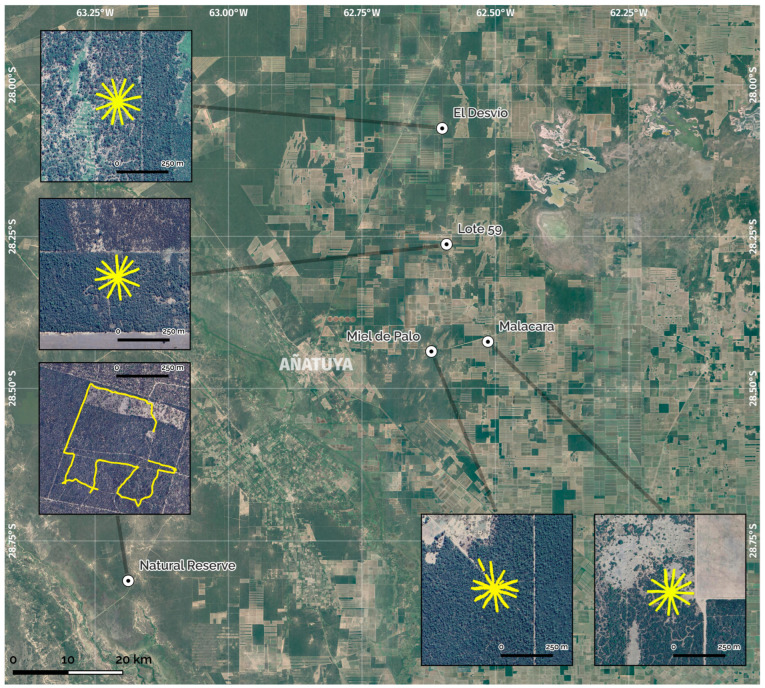
Grid-based transects applied in each of the study areas according to Parmenter, R.R. et al., 2003 [17]. For the ANR, capture stations were installed steps away from pre-existing paths.

**Figure 3 insects-15-00471-f003:**
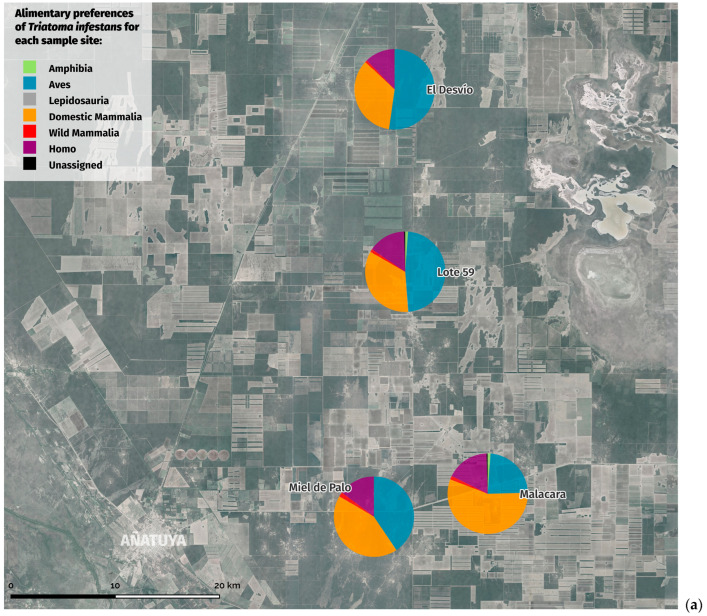
Results obtained on the bloodmeal preferences of each of the species captured in each of the study sites: (**a**) *T. infestans*; (**b**) *T. garciabesi;* and (**c**) *T. guasayana*.

**Figure 4 insects-15-00471-f004:**
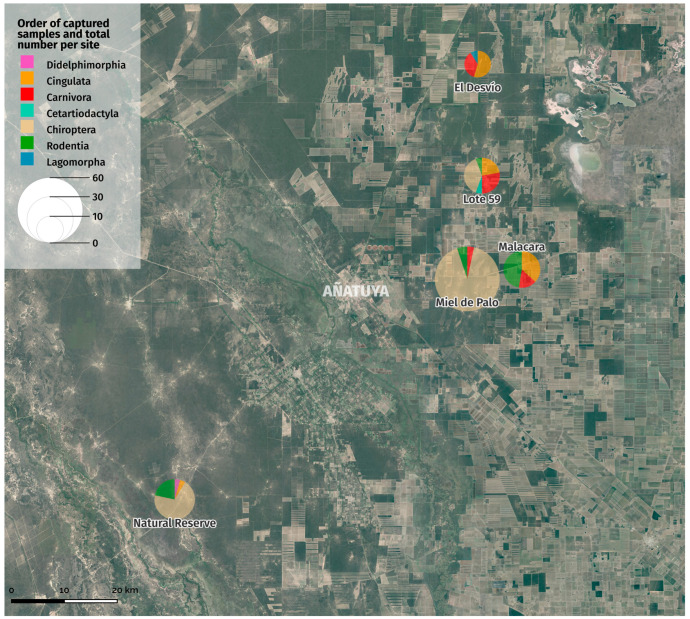
Summary of the species richness, belonging to the main orders of wild animals captured in each of the study sites.

**Figure 5 insects-15-00471-f005:**
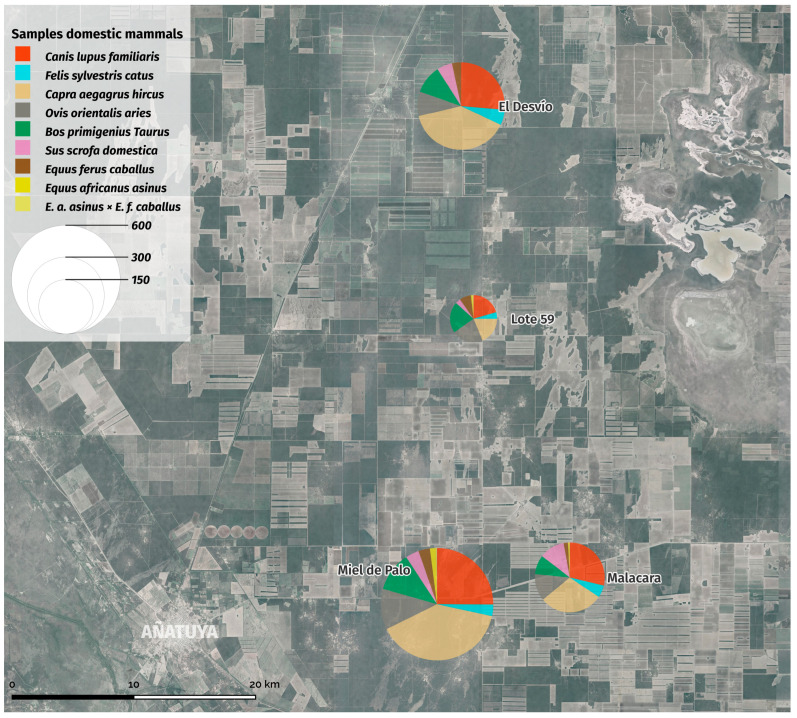
Summary of the species richness of domestic animals captured in each of the study sites.

**Table 1 insects-15-00471-t001:** Control and surveillance activities were carried out at the study sites with different degrees of intensity and times, including insecticide spraying, sanitary improvement of human habitat, and diagnostic and treatment operations.

Sampling Site	Number of Surveillance and Control Cycles	Sanitary Improvement of Human Habitat	Diagnostic and Treatment Operations
ED	Start in 2005 Total of cycles: 17	Done Between 2005 and 2007	Done 2017–2019
MP	Start in 2006 Total of cycles: 18	Done Between 2005 and 2006	Done 2016–2019
L59	Start in 2016 Total of cycles: 4	Done Between 2018 and 2018	Done In 2019
M	Start in 2019 Total of cycles: 1	In process	Done In 2019

**Table 2 insects-15-00471-t002:** Number of captured wild mammals in the study sites from General Taboada and Ibarra departments, Santiago del Estero province, Argentina (2017–2019). The nomenclature follows the proposal of Teta et al., (2021) [41], with the addition of some subsequent changes [42].

Order/Species	Study Sites					Total Number of Specimens from Each Captured Species
	Miel de Palo	Malacara	Lote 59	El Desvío	Natural Reserve	
Didelphimorphia						
*Thylamys pulchellus*					1	1
Cingulata						
*Chaetophractus vellerosus*		1				1
*Chaetophractus villosus*		4	4	3	1	12
*Tolypeutes matacus*		1		3		4
*Dasypus hybridus*		1				1
Carnivora						
*Leopardus geoffroyi*			1			1
*Lycalopex gymnocercus*	1	1		1		3
*Conepatus chinga*	1	2	4	3		10
Cetartiodactyla						
*Mazama gouazoubira*			1			1
Chiroptera						
*Desmodus rotundus*			7			7
*Tadarida brasiliensis*					1	1
*Promops nasutus*	18					18
*Eptesicus* cf. *furinalis*	24					24
*Myotis* spp.	12				15	27
Rodentia						
*Graomys chacoensis*	1	1				2
*Akodon toba*		3			4	7
*Rattus rattus*	1					1
*Galea leucoblephara*		1	1		1	3
*Lagostomus maximus*	1	4				5
Lagomorpha						
*Lepus europaeus*				1		1
Total number of captured species per site	8	10	6	5	6	-
Total number of captured samples per site	59	19	18	10	23	129

**Table 3 insects-15-00471-t003:** Number of domestic mammals sampled in the study areas from General Taboada and Ibarra departments, Santiago del Estero province, Argentina (2017–2019).

Species (% of Census Population Sampled)	Study Sites				Total Number of Specimens from Each Species
	Miel de Palo	Malacara	Lote 59	El Desvío	
*Canis lupus familiaris* (100%)	161	71	23	102	357
*Felis sylvestris catus* (50%)	21	14	5	22	63
*Capra aegagrus hircus* (10%)	252	73	21	152	498
*Ovis orientalis aries* (20%)	75	31	24	35	165
*Bos primigenius Taurus* (25%)	73	22	24	41	160
*Sus scrofa domestica* (50%)	25	29	4	22	80
*Equus ferus caballus* (25%)	21	5	9	13	48
*Equus africanus asinus* (25%)	11	1	1	0	13
*E. a. asinus × E. f. caballus* (50%)	2	1	1	0	4
Total of sampled animals per site	641	247	112	387	1388

## Data Availability

Data is contained within the article and Appendix A.

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
