# Peer review of "Zoonotic Cycle of American Trypanosomiasis in an Endemic Region of the Argentine Chaco, Factors That Influenced a Paradigm Shift"

_insects, 2024, doi:10.3390/insects15070471_

Round 1

Reviewer 1 Report

Comments and Suggestions for Authors

Rev. Gomez-Bravo et al.

This report of the distribution of T. cruzi infection in wild and domestic mammals and triatomines in the Chaco provides new and valuable information for a topic of increasing global relevance, pathogen spillover. The authors of this paper include prestigious researchers and leaders in this field. Therefore, I believe this paper will be of interest to a broad readership after some revisions are undertaken. I provide constructive comments with this goal in mind.

                A key issue is whether a one-year survey of wild or domestic mammals and triatomines can provide sufficient evidence for assessing transmission dynamics, pathogen spillover and the various impacts of landscape change (deforestation and so on). I don’t think so. Consequently, some of the statements need to be revised and toned down. The absence of evidence before said changes severely hampers the value of any inference.

                The second chapter to consider for strengthening this paper is data reporting (see below). The key data are hidden in the text and the Abstract is quite uninformative.

                There are few aspect that spurred my curiosity. The paper does neither mention the occurrence of Didelphis opossums in the area nor does it mention that they are the most widespread reservoir host of T. cruzi in the Chaco and throughout the Americas. The paper should comment on this singularity and add some references for this specific region. You may also compare your results with those from other studies in the region.

                If the mammal samples reported in this paper also appeared in a closely related paper by Wehrendt et al. (2019, in the reference list), you’d say so to avoid duplication in any future meta-analysis of the subject.

Specific comments

Abstract

This needs to be rewritten in full. Half of the Abstract conveys introductory information, and no method, specific result and conclusion is given, just the objectives of the study. The latter “aims to describe, in an area with human intervention and surveillance and vector control, the transmission dynamics of the sylvatic cycle of T. cruzi, identify factors determining potential zoonotic spillover…”.

Introduction

l. 55, I believe not all triatomine species have been shown to be infected with T. cruzi, though they may be potentially susceptible to the parasite.

l. 81-2, “interruption…” of parasite transmission is based on vector control and other types of screening for T. cruzi infection (transfusion, pregnant women).

l. 84-6, your statement may well be true, though please note that many areas have been certified “free of transmission”, so is it there an inconsistency or is it that there is a lack of information for certain regions or settings?

l. 87-9, the sentence omits house infestation with triatomines. You may briefly identify what type of control interventions are available.

l. 117, “vector control activities…”

l. 116-8, This is a key issue affecting the ability of any study to untangle the dynamics of transmission and identifying factors linked to zoonotic spillover (main aims of this study): Is there any information on the features of the sylvatic transmission cycle of T. cruzi in this area in the past by the time vector control actions were initiated? If you you’d provide references on this regard. If there is no such information, they you’d acknowledge its absence and limit the inferences this study could make.

Methods

Figure 1. Provide more details on what is shown.

Table 1: readers probably are not aware what means “control cycles”, please define. By the same token, define “diagnostic and treatment operations”. Is this related to house infestation or human treatment?

l. 188, how did you choose where to place the transects?

l. 191, improve wording. It seems you’re referring to the 12 transects, but in this case you’re only referring to 10 of them.

Figure 2: revise reference. I cannot see the 12 transects in this Figure, are they there?

l. 217-8: unify both sentences, not a full stop before While.

l. 219-20, Is capture effort equal to net length divided by time? Please provide a reference. I would think it is a composite measure, net length x time.

l. 278: revise wording.

Section 2.5: when were domestic animals censused and tested?

Results

Table 2: this is too lengthy for just seven columns. Mammal orders should not cut across the body of the table; they should fit under Order/species.

The same issues apply to Table 3.

Figure 4: this is too much space for very little information that hardly can be seen. You may consider using a supplementary table.

Sections on Triatomines and Detection of T. cruzi (for triatomines): these data should be displayed in a table by species and type of habitat (domestic, peridomestic, sylvatic) where readers can in a quick glance get the message of distribution and infection (lines 425-35). Line 435 seems incomplete.

Section 3.5 on bloodmeal results: this should be in a table by village, species and host instead of a Figure 5, which hardly show the data in a clear way.

Section 3.4 should be moved after the data on triatomine distribution and infection. I also believe you’d show your results in a table accommodating techniques, operating labs and main outcomes. These may be the core results of this paper and they’re quite hidden in the text. For example, you have discordant results for the wild mammals, and also discordant results between molecular and serological techniques. Is this adequately covered in the Discussion?

Section 3.6 on human seroprevalence of T. cruzi appears here quite unexpectedly because it had not been mentioned among the study aims and methods. Ethical issues are key concerns here. Human samples come from areas without “risk of transmission”: how do you define this? How was the selection of human subjects conducted? You may either expand this section after providing all the necessary items, or delete it altogether and or mention these data as unpublished data and the source in the discussion section.

Discussion

l. 470-3 : you cannot really tell if there was a decrease of infection in wild mammals because you have shown no previous data. Similarly, you cannot impute a negative effect on the several factors you mention unless you provide some reference to work elsewhere or show evidence in this paper.  This issue appears again in l. 553-5.

l. 474-6: you’d reconcile this statement with your study aims at the end of the introduction.

I don’t see how you can “identify the specific sites affected by human intervention”.

l. 480-1: from the text in Section 3.4, I understook that one of the labs identified T. cruzi in quite a fraction of the samples, or are these results based on kDNA held invalid (re discussion l. 503-?

l. 543-550: You’d provide references to support your conclusions. Your prediction that T. guasayana is responsible of dog infections should be toned down without losing substance: the evidence provided is simply not enough.

References: revise formats thoroughly.

Comments on the Quality of English Language

Need to be polished more.

Author Response

We appreciate the careful review carried out by the reviewer.

  • As suggested, the Abstract has been rewritten in full. Many thanks for the suggestion:

    Abstract: Trypanosoma cruzi, the causative agent of Chagas disease (American trypanosomiasis) is a highly complex zoonosis, that is present throughout South America, Central America, and Mexico. The transmission of this disease is influenced by various factors, including human activities like deforestation and land use changes, which may have altered the natural transmission cycles and their connection to the environment. In this study conducted in the Argentine Chaco region, we examined the transmission dynamics of T. cruzi by collecting blood samples from wild and domestic animals, as well as triatomine bugs from human dwellings, across five sites of varying anthropic intervention. Samples were analyzed for T. cruzi infection via qPCR, and we additionally examined triatomines for bloodmeal analysis via NGS amplicon sequencing. Our analysis revealed a 15.3% infection rate among 20 wild species (n=123) and no T. cruzi presence in 9 species of domestic animal (n=1359) or collected triatomines via qPCR. Additionally, we found chicken (34.28%), human (21.59%), and goat (19.36%) as the predominant bloodmeal sources across all sites. These findings suggest that anthropic intervention and other variables analyzed may have directly impacted the spillover dynamics of T. cruzi's sylvatic cycle and potentially reduced its prevalence in human habitats.

  •   In reference to the occurrence of opossums,the following paragraph and references has been included in the text : Didelphis albiventris is one of the most widely distributed marsupials in Argentina (Chemisquy and Martin, 2019). In the Chaco region, it has been identified as a major reservoir host of Trypanosoma cruzi (   ). Despite its high adaptability to various environmental disturbances and significant sampling efforts, the species was not recorded in the study area. This absence is likely due to the low population density and small size of forest patches, which negatively impact the presence of the opossum as it relies on trees for nesting. Furthermore, indiscriminate hunting in rural areas, where it is perceived as a predator of poultry, exacerbates the decline in its population. Alvarado-Otegui, J. A., Ceballos, L. A., Orozco, M. M., Enriquez, G. F., Cardinal, M. V., Cura, C., ... & Gürtler, R. E. (2012). The sylvatic transmission cycle of Trypanosoma cruzi in a rural area in the humid Chaco of Argentina. Acta Tropica, 124(1): 79-86.

Chemisquy, M. A., Martin, G. M. (2019). Didelphis albiventris. En: SAyDS–SAREM (eds.) Categorización 2019 de los mamíferos de Argentina según su riesgo de extinción. Lista Roja de los mamíferos de Argentina. Versión digital: http://cma.sarem.org.ar.

Orozco, M. M., Enriquez, G. F., Alvarado-Otegui, J. A., Cardinal, M. V., Schijman, A. G., Kitron, U., & Gürtler, R. E. (2013). New sylvatic hosts of Trypanosoma cruzi and their reservoir competence in the humid Chaco of Argentina: a longitudinal study. The American Journal of Tropical Medicine and Hygiene, 88(5): 872–882.

Orozco, M. M., Enriquez, G. F., Cardinal, M. V., Piccinali, R. V., & Gürtler, R. E. (2016). A comparative study of Trypanosoma cruzi infection in sylvatic mammals from a protected and a disturbed area in the Argentine Chaco. Acta Tropica, 155: 34-42.

Rabinovich, J. E., Schweigmann, N., Yohai, V., & Wisnivesky Colli, C. (2001). Probability of Trypanosoma cruzi transmission by Triatoma infestans (Hemiptera: Reduviidae) to the opossum Didelphis albiventris (Marsupialia: Didelphidae). American Journal of Tropical Medicine and Hygiene, 65(2): 125-130.

  • I. 55. ... corrected: the vast majority
  • I.82 2 ...paragraph deleted. Thanks for the observation!
  • I.87 9 ...done
  •  116-8, This is a key issue affecting the ability of any study to untangle the dynamics of transmission and identifying factors linked to zoonotic spillover (main aims of this study): Is there any information on the features of the sylvatic transmission cycle of T. cruzi in this area in the past by the time vector control actions were initiated? The following paragraph has been added:
  • In a study carried out in the Gran Chaco region, Ceballos et al. [ ]  compared the variation in the transmission cycle of T. cruzi between surveys carried out 1984 - 1991 and 2002-2004, and  observed a significant decrease in the prevalence and incidence over time. In 2002 and 2004, 501 samples of wild mammals from 13 different species were analyzed, and only 2 species were found infected by T. cruzi, an opossum Didelphis albiventris and a skunk Conepatus chinga. These results are consistent with those found by Alvarado-Otegui et al.  [ ]   , in which 44 samples of mammals belonging to 14 species were analyzed, of which only 2 opossums Didelphis albiventris and one armadillo Dasypus novemcinctus were infected with T.cruzi. In our study Didelphis albiventris was not reported in any of the study sites,although the presence of skunk and armadillo species was reported, none were infected by T. cruzi.

  • Section 2.5: when were domestic animals censused and tested?: Domestic animals were censused in 2017 and sampled between 2017 and 2018.
  • Tables and figures: We believe they are appropriate and fit the format of the journal.
  • Section 3.6 on human seroprevalence of T. cruzi appears here quite unexpectedly because it had not been mentioned among the study aims and methods.... Section 3.6 belongs to the methods section:

    Since 2017, the local health system, in collaboration with Mundo Sano, has been carrying out serological screening on Chagas disease, in rural areas without risk of vector transmission. At the sites included in this study (Table 1), between 2017 and 2019, 202 persons were screened (31% coverage) and 68 (34%) were positive with conventional serological procedures: 1 (1,5%) children under 5 years, 1 (1,5%) children between 6 and 10 years, 1 (1,5%) between 11 and 30 years, and 62 adults up to 31 (unpublished data).

  • l. 470-3  References were added and discussed in the Discussion section:

    Ceballos LA, Cardinal MV, Vazquez-Prokopec GM, Lauricella MA, Orozco MM, Cortinas R, Schijman AG, Levin MJ, Kitron U, Gürtler RE. Long-term reduction of Trypanosoma cruzi infection in sylvatic mammals following deforestation and sustained vector surveillance in northwestern Argentina. Acta Trop. 2006 Jul;98(3):286-96. doi: 10.1016/ j.actatropica.2006.06.003. Epub 2006 Jul 12. PMID: 16839513; PMCID: PMC1853287.

    Alvarado-Otegui JA, Ceballos LA, Orozco MM, Enriquez GF, Cardinal MV, Cura C, Schijman AG, Kitron U, Gürtler RE. The sylvatic transmission cycle of Trypanosoma cruzi in a rural area in the humid Chaco of Argentina. Acta Trop. 2012 Oct;124(1):79-86. doi: 10.1016/j.actatropica.2012.06.010. Epub 2012 Jul 3. PMID: 22771688; PMCID: PMC3444808.

  • l. 474-6  Done
  • l. 543-550 Done

English language has been revised.

References have been revised.

Reviewer 2 Report

Comments and Suggestions for Authors

This is an interesting article which investigates the prevalence of Trypanosoma parasites in wild and domestic animals in an endemic region of the Argentine Chaco. There is a control plan for the disease in place, and it appears to be working quite well given that they are largely negative results which are reported.

This doesn’t take away from this being a nice, and interesting paper which adds details to the field.

I only have a few minor comments which are below.

It maybe worth giving the text a proof read to polish the text – it is slightly awkwardly written in places and there are some typos and grammatical errors present.

Table 1 in MP- is this 2019 rather than 1019?

Line 237- was the sampling here the same as for wild animals- e.g. anaesthesia and then blood sampling? Perhaps a bit more detail needed in here

Line 257- please include details on negative and positive samples

Can you confirm PCR regents and cycling conditions were as per the previous study in the text please?

Line 278-  is there a word missing here- traps and? ?

Can you please include manufacturers for reagents in the methodology?

Figure 3 and 4 and 5- im not sure if it is my copy, but the graphs are very unclear and feint, Is it possible to make them brighter please?

Line 403- highest is a typo

Line 435- this sentence appears unfinished ?

Section 3.6. This is not included in the methods- please can you include some brief details of this in the methods?

Comments on the Quality of English Language

These are detailed above

Author Response

We appreciate very much the reviewer's comments and suggestions.

We highly appreciate the reviewer´s comments

  • It maybe worth giving the text a proof read to polish the text... Text has been revised for English language.
  • Table 1 in MP- is this 2019 rather than 1019? ...Corrected. Many thanks!
  • Line 237- was the sampling here the same as for wild animals- e.g. anaesthesia and then blood sampling? Perhaps a bit more detail needed in here..paragraph was modified: Blood samples collected from both wild and domestic mammals were kept in guanidine hydrochloride-EDTA buffer (GEB) in a 3:1 ratio at room temperature in the field laboratory until they were sent to the Instituto de Investigaciones en Ingeniería Genética y Biología Molecular (INGEBI) in Buenos Aires (Argentina) for analysis to determine the presence of T. cruzi
  • Line 257- please include details on negative and positive samples.  Control samples were obtained from a previous study:Turriago Gómez, B.C.; Vallejo, G.A.; Felipe, G. SEROPREVALENCIA DE Trypanosoma cruzi EN PERROS DE DOS ÁREAS ENDÉMICAS DE COLOMBIA. Revista Med 2008, 16, 11–18.
  • Line 278-  is there a word missing here- traps and? Paragraph corrected: ...  During the summer sampling, traps were installed in the peri-domicile, in structures such as goat pens....
  • Figure 3 and 4 and 5- im not sure if it is my copy, but the graphs are very unclear and feint, Is it possible to make them brighter please? Thanks for comment.We will contact the Editor
  • Line 403- highest is a typo.Corrected .Thanks!
  • Line 435- this sentence appears unfinished ? Indeed, many thanks! Paragraph was corrected:

    ...95 were also analyzed for traces of T. cruzi DNA and there was no detection of parasites.

    Section 3.6. This is not included in the methods- please can you include some brief details of this in the methods? 

    We obtained the information via personal communication (unpublished data).

  • Englis language and references have been revised

Reviewer 3 Report

Comments and Suggestions for Authors

The manuscript under title (Zoonotic cycle of American trypanosomiasis in an endemic region of the Argentine Chaco, factors that influenced a paradigm shift) investigated the effect human interventions; deforestation, largescale changes in land use, and loss of biodiversity, along with sustained control interventions, may have significantly impacted the structure of wild transmission cycles of T. cruzii and their relationships with the environment. The study is an interesting work and well organized to manage large data of this work. There some points needs clarification from the authors:

 Materials and Methods

The authors could add another natural photos contain the environment components in (Figure 1. Study area

Table 1 contains abbreviations like ED, MP, M, L59; please add complete name in foot note of table

2.6. Molecular diagnosis in mammals: please add molecular diagnosis of T. cruzii

2.8. Molecular diagnosis in triatomine insects: please add T. cruzii

Results:

Wild animals:how did the authors identified all these animals?

The authors can supplied the common name of animals beside the scientific one

Triatomines, did the authors found more than species in the same site?

Fig. 3, 4, and 5, need to change the back ground to easily to read

Discussion

The authors mentioned the following paragraph 551 to 555; was mentioned without reference. Also, the conclusion of this is contradicted with the presence of Triatomines

The authors need to discuss the finding in relation to the sampling sites. Also, there is no significant difference between the three sites why?

Author Response

We highly appreciate the reviewer´s comments.

  • Table1 We have used the abbreviations that appear in the text.
    The names of the places are very long and that is why we use
    abbreviations that are sufficiently clarified in the body of the manuscript.
  • 2.6 Done
  • 2.8 Done
  • Fig 3,4,5 We will comment to the editor
  • 551-555  Done
  • Discussion has been improved 

Round 2

Reviewer 1 Report

Comments and Suggestions for Authors

The authors have improved the text in many respects. I  believe the Tables and Figures in its present form do not convey clear messages and can be improved with some extra work. In any case, the Editor should decide whether these formats are appropriate.

I picked one typo: agroFcals.

Author Response

Many thanks to the reviewer for further comments. We have carefully reviewed the text and incorporated suggestions and comments.
In fact, we are going to contact the editor to decide the best format for
the tables and figures.
The  typo was corrected.